# Stable de novo protein design via joint conformational landscape and sequence optimization

Yehlin Cho[1], Justas Dauparas [2], Kotaro Tsuboyama[3,4], Gabriel J. Rocklin [3] ✉ & Sergey Ovchinnikov [1] ✉

Generative protein modeling provides advanced tools for designing diverse protein sequences and structures. However, accurately modeling the conformational landscape and designing sequences remain critical challenges: ensuring that the designed sequence reliably folds into the target structure as its most stable conformation, and optimizing the sequence for a given suboptimal fixed input structure. In this study, we present a systematic analysis of jointly optimizing sequence-to-structure and structure-to-sequence mappings. This approach enables us to find optimal solutions for modeling the conformational landscape. We validate our approach with large-scale protein stability measurements, demonstrating that joint optimization is superior for designing stable proteins using a joint model (TrRosetta and TrMRF) and for achieving high accuracy in stability prediction when jointly modeling (half-masked ESMFold pLDDT + ESM2 Pseudo-likelihood). We further investigate features of sequences generated from the joint model and find that they exhibit higher frequencies of hydrophilic interactions, which may help maintain both secondary structure registry and pairing-features not captured by structure-to-sequence modeling alone.

Generative modeling is becoming increasingly important in protein design, enabling the creation of diverse de novo structures with sequences that can refold into desired conformations[1–4]. Stability must be a top priority when designing proteins, as it enhances the efficiency of protein expression, purification, and crystallization[5]. Stability is primarily determined by a balance of forces that govern the free energies between the folded and unfolded states of the protein. It results from the intricate interplay between atomic interactions in the folded state and the higher conformational entropy in the unfolded state. Unstable proteins are prone to degradation and aggregation, making them unsuitable for biological or therapeutic applications. In an unfolded state, proteins can lose their function and potentially cause diseases[6,7].

One primary approach to designing a folded protein is to generate a sequence that can fold into a specified target structure using structure-to-sequence models. These models ($P$(sequence|structure)) are often called "Inverse Folding" models, as they reverse the sequence-to-structure prediction process. However, many models labeled as such do not achieve true inverse folding. True inverse folding must satisfy two conditions: (i) the sequence must fold into the designed structure as its most energetically favorable configuration, and (ii) the sequence must not fold into any alternative structure with the same free energy[8]. A model trained with a $P$(sequence|structure) objective cannot see alternative conformations besides the one given, so there is a chance that the design sequences fold into lower energy conformations other than the target structure. As illustrated in Supplementary Fig. 1, $P$(sequence|structure) models can fail by producing sequences that result in structures with swapped strands or disrupted backbone interactions, thereby folding into alternative structures.

[1]Massachusetts Institute of Technology, Cambridge, MA, USA. [2]Institute for Protein Design, University of Washington, Seattle, WA, USA. [3]Department of Pharmacology & Center for Synthetic Biology, Northwestern University Feinberg School of Medicine, Chicago, IL, USA. [4]Institute of Industrial Science, The University of Tokyo, Tokyo, Japan. ✉e-mail: grocklin@gmail.com; so3@mit.edu

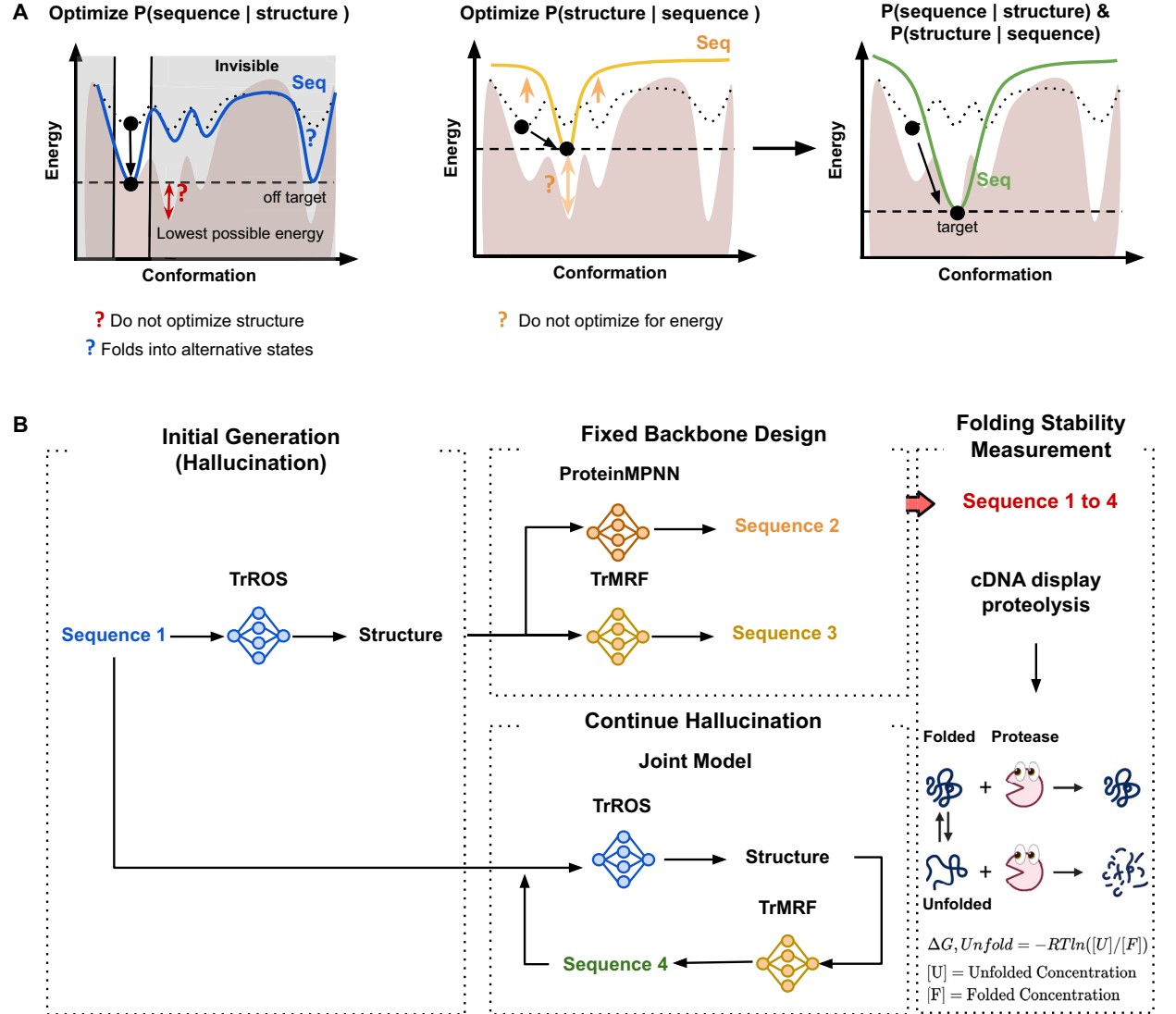

**Fig. 1 | Overview of joint optimization in sequence and structure models. A** Left The structure-to-sequence model optimizes $P$(sequence|structure), designing a sequence based on a fixed backbone structure. However, it cannot refine the structure itself, and the designed sequence may adopt alternative conformations. **A**, Middle The sequence-to-structure model optimizes $P$(structure|sequence), redesigning a sequence based on the output structure, smoothing the conformational landscape and adjusting it toward the optimal conformation by maximizing its probability relative to other states. However, once the target conformation is optimized, there is no further driving force to optimize the sequence for a lower energy minimum compared to alternative conformations. **A** Right A joint model integrates both sequence and structure information, better modeling the conformational landscape. The brown-filled landscape represents the lowest possible energy for different conformations. The curved line represents a sampled sequence, the black-filled circle indicates a conformation, and the dotted curved line shows the sequence from the previous step. **B** End-to-end sequence generation is performed using four methods-TrROS (TrRosetta), TrMRF (transform-restrained Markov Random Field), ProteinMPNN, and the Joint model (TrROS + TrMRF). After generating the sequences, protein stability was measured using the cDNA display proteolysis method.

Another problem is that, since the model is a fixed-backbone sequence design model, it inherently assumes that the input structures are good and does not optimize for $P$(structure) when designing a sequence, potentially limiting the ability to further lower the energy due to structural constraints (Fig. 1A, Left).

Structure prediction models ($P$(structure|sequence)), which are models finding the most likely structure for a given protein sequence can be viewed as models that can see the entire energy landscape for a given sequence (Fig. 1A, Middle). These models can be used for sequence design through gradient backpropagation, identifying sequences that maximize the probability of the structure relative to all other possible states[9]. This approach has the potential to address the problems of alternative states and fixed backbone design in the structure-to-sequence

model. However, most structure prediction models are trained on valid sequences known to fold, so the model inherently assumes that the sequences are good (i.e., have a high $P$(sequence)). As a result, the model may still consider a sequence as a good sequence, even if it is adversarial and does not fold well. For example, $P$(structure|sequence) models often struggle to differentiate between stable and unstable protein designs[10] and frequently fail when applied to protein design tasks alone[11,12]. In addition, TrRosetta (TrROS), an example of a $P$(structure|sequence) model, is effective at disfavoring alternative conformational states but is relatively limited in its ability to reduce the absolute energy of a design compared to energy-based models such as Rosetta[9].

We hypothesize that jointly optimizing models that predict both $P$(sequence|structure) and $P$(structure|sequence) will enable us to

model the conformational landscape by optimizing the structure to achieve the lowest possible energy. By exploring the full energy landscape, the joint model can prevent the sequence from folding into alternative states, allowing the backbone to be adjusted to accommodate the best possible sequence, thereby achieving a global minimum energy state (Fig. 1A, Right).

Here we provide a systematic analysis of (1) $P$(sequence|structure) (2) $P$(structure|sequence), and (3) the joint optimization of $P$(sequence|structure) and $P$(structure|sequence) using a four-model protocol, where each model takes one of (1)–(3) as its objective. The four models are designed to generate four sequences that share the same fold but differ in sequence, optimizing based on different objectives. We experimentally measure the folding stability of all designs using a cDNA display-based proteolysis method[13], with stability serving as an indicator of how well each model, employing different objective functions, is modeling the conformational landscape and design sequences. This work presents a large-scale, head-to-head comparison of different design methods, comprising 13,442 stability measurements. It thus provides an opportunity to evaluate how well current models with varying objectives understand protein stability. We also hypothesize that jointly optimizing both sequence and structure models will perform better at predicting and ranking stable proteins than using a single objective model. We approximate the confidence and likelihood scores of existing large protein models as zero-shot predictors of stability and evaluate how well they correlate with our stability measurements. Our work highlights the importance of joint optimization of sequence and structure in modeling the conformational landscape to generate and score stable proteins.

## Results

### Generating diverse protein sets by jointly optimizing sequence and structure

We investigate four models to generate de novo protein structures and sequences: TrROS, TrMRF, joint (TrROS + TrMRF), and ProteinMPNN (Fig. 1 D). TrROS[9] is a model that generates the protein structure given a sequence ($p$(structure|sequence)). For unconditional protein generation to create thousands of structures and sequences, we use TrROS, starting with random amino acids and continuously applying gradient backpropagation until a clear distance matrix is obtained[9,14]. For the structure-to-sequence task, we used both ProteinMPNN and our newly trained TrMRF model to generate sequences from the predicted structures. The workflow involves: (1) generating structures with TrROS, (2) generating sequences from these structures using ProteinMPNN/TrMRF, and (3) applying joint optimization to refine both sequences and structures (details in Supplementary Fig. 2). To ensure compatibility with downstream experimental validation, we specifically designed mini-proteins containing fewer than 80 residues, allowing stability measurements via the cDNA display method within a high-confidence range. Cysteine is excluded to prevent the formation of disulfide bonds.

As a result, for each structure, we generated one sequence for each of the four different models. In total we obtained and analyzed the stability measurement of 20,668 sequences (5167 per model). After filtering the sequences based on low AlphaFold2 inter-PAE (inter-chain predicted alignment error) to exclude proteins with potential aggregation or homo-oligomer formation, we retained 13,442 sequences, of which 5708 (1427 per model) have sequences from all four models. As shown in Supplementary Fig. 5, 54% are classified as $\beta$, 32% as $\alpha + \beta$, and 8% as $\alpha$, indicating a predominance of $\beta$ containing structures.

Sequences with the same structures show low similarity across different models. The joint model and ProteinMPNN have the lowest average sequence similarity at 0.25, while TrROS and TrMRF have the highest at 0.407. The average sequence similarity between TrROS and the joint model is 0.402, indicating that further optimization, especially when combined with the TrMRF model, can generate sequences

that differ significantly from those produced by TrROS alone. This also supports the observation that sequence design models can often generate multiple diverse sequences that fold into the same target structure, as demonstrated in previous studies[15,16]. To confirm whether sequences from the joint model, which continue hallucinating from the output sequence of TrROS, are altering the backbone into a completely different fold, we compared the differences in structure between sequences from the joint model and those from other models using RMSD (root mean square deviation). While there are no large-scale structural differences between the joint model and the others-comparable to the variability observed among the other models themselves (Supplementary Fig. 3), the RMSD distributions for the joint model are slightly shifted to the right (Supplementary Fig. 3), indicating modest structural perturbations.

### Joint optimization generates the most stable proteins by modeling the conformational landscape

We performed experimental validation by measuring protein folding stability using a high-throughput cDNA display proteolysis-based method[13,17–19]. Further details are provided in the Supplementary Information. As shown in Fig. 2A, the sequences generated by the joint model have the highest $\Delta G_{,\mathrm{Unfold}}$ (which has the opposite sign of $\Delta G_{,\mathrm{Fold}}$), representing the highest stability according to their highest resistance to two proteases, trypsin and chymotrypsin. To analyze the effect of sequence on folding stability, we compared 5708 sequences (1427 per model) from different models that share nearly identical structures and calculated the difference in folding stability between the joint model sequences and those from other models. Sequences generated by the joint model (Fig. 2B) exhibit higher folding stability than those from TrROS, TrMRF, and ProteinMPNN, with percentages of 80.5%, 74.4%, and 84.7%, respectively. This indicates that sequences designed by the joint model are more stable within the same fold.

A common approach in protein design involves using a structure-to-sequence model to generate protein sequences and filtering these designed sequences based on computational confidence metrics, such as the AlphaFold2 (AF2) pLDDT score, which confirms that the designed proteins are well-folded with high confidence. For sequences generated from structure-to-sequence models (TrMRF and ProteinMPNN), we applied a filter of AF2 pLDDT > 85, where the AF2 prediction is based on the default setting of single-sequence prediction with three recycling steps. As shown in Fig. 2C, the filtered sequences from the TrMRF and ProteinMPNN models exhibit a folding stability distribution similar to that of the joint model sequences without filtering. TrMRF and ProteinMPNN, after applying the AF2 pLDDT > 85 filter, retain 237 out of 2854 original sequences with a median stability of 3.08 kcal mol$^{-1}$, whereas the joint model without filtering shows a median stability of 2.90 kcal mol$^{-1}$. This is because, theoretically, designing a sequence based on a structure and then confirming its conformation using a structure prediction model can achieve the same objectives as the joint model, optimizing both $P$(structure|sequence) and $P$(sequence|structure). However, filtering joint model sequences with AF2 pLDDT > 85 shows an additional shift toward higher stability, retaining 12% of sequences with a median stability of 3.66 kcal mol$^{-1}$.

As shown in Supplementary Fig. 8, the adjusted structures from the joint model exhibit lower ProteinMPNN entropy, when compared to the structures from the initial generation by the TrROS model, which were used as input for both the TrMRF and ProteinMPNN models. This supports our explanation that joint optimization helps the model adjust the structure to achieve a higher $P$(structure). This highlights that the joint model, by continuously adapting to both structural and sequential changes, may outperform single-objective models in designing highly stable structures, even after filtering with commonly used computational standards.

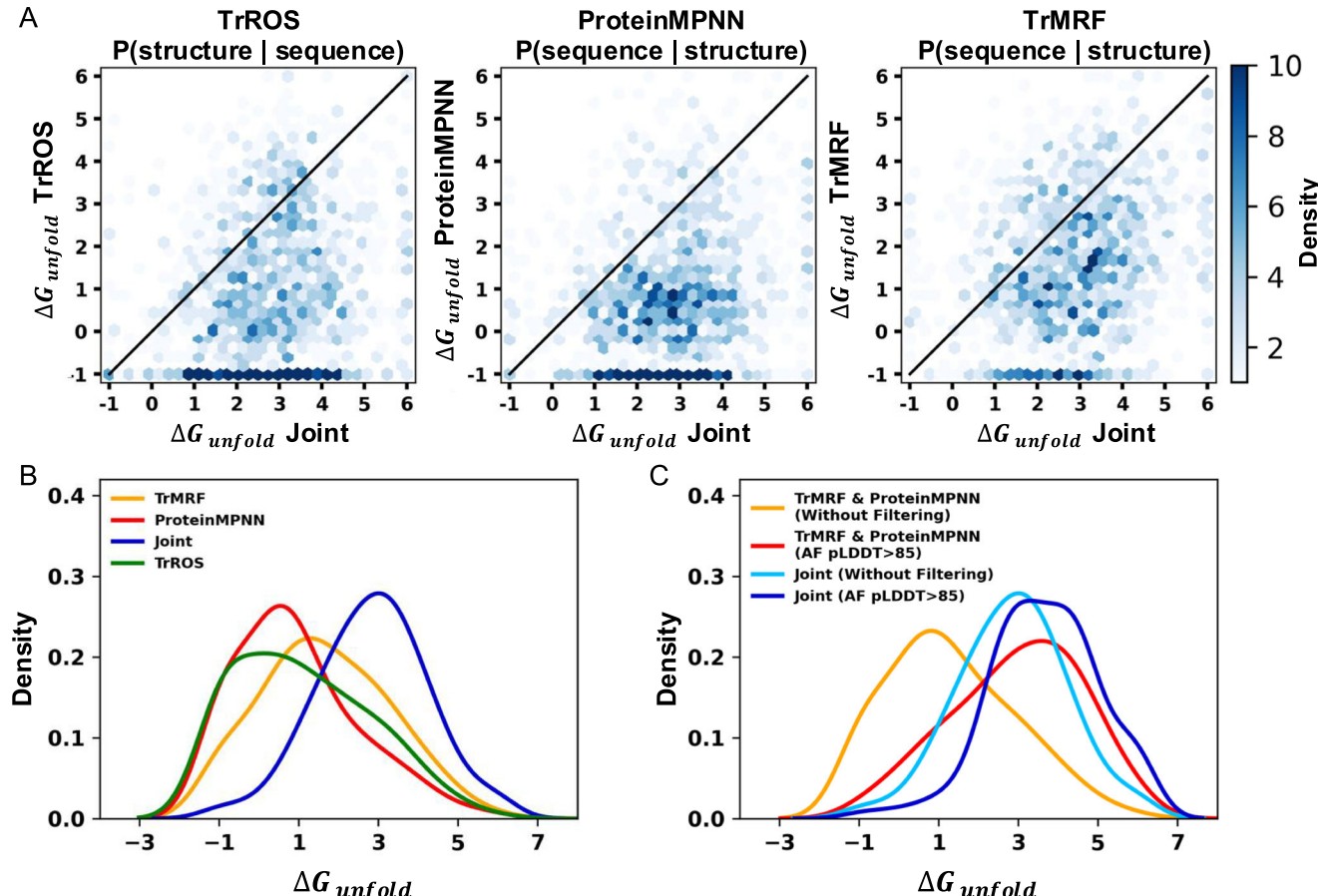

**Fig. 2 | cDNA display proteolysis-based folding stability measurements show that the joint model generates more stable designs. A** Comparison of $\Delta G_{\text{unfold}}$, shown in a hexaplot with the joint model on the x-axis and the single model on the y-axis, comparing the $\Delta G_{\text{unfold}}$ of sequences from each model that share the same structure. **B** Distribution of $\Delta G_{\text{unfold}}$ of four models. **C** Distributions of $\Delta G_{\text{unfold}}$ for sequences generated by (1) the joint model and (2) TrMRF and ProteinMPNN, with and without filtering for AF2 pLDDT > 85.

## Large sequence and structure models score folding stability in a zero-shot setting

Although filtering sequences based on common computational standards shifts the distribution toward more stable sequences compared to unfiltered ones, highly stable designs are not always achieved through common computational filtering (Fig. 3A). Notably, among all sequences with high folding stability ($\Delta G > 5$), only 21.7% passed the AF2 (pLDDT > 85) and ProteinMPNN (Cross Entropy (CE) < 1.5) filters. These thresholds reflect widely used design criteria, where CE < 1.5 corresponds to average ProteinMPNN perplexity, and pLDDT > 85 is commonly used to filter low-confidence regions[20,21]. A significant number of sequences exhibit moderate confidence values but high folding stability. Many highly stable designs still fall outside the filtering criteria, and given the heterogeneity of our dataset, we have the opportunity to explore alternative computational scoring matrices that may better correlate with folding stability. We assess how well current protein models rank stable versus unstable designs.

Large protein models trained without fitness labels-such as activity, stability, or expression-have demonstrated the ability to predict and rank fitness metrics[22]. This capability may arise from the diverse, naturally occurring protein training datasets relevant to biological functions[23]. Zero-shot prediction enables the evaluation of a model's ability to predict new tasks and classes without additional training[24,25]. Similarly, we obtain model confidence and likelihood scores in a zero-shot setting and correlate them with actual stability scores to assess model performance. For sequence-conditioned models, we use the protein language model ESM-2[26], ESMFold[27], and

AF2. For structure-conditioned models, we use ESM-IF[28,29] and ProteinMPNN[15]. We evaluate their ability to predict protein folding stability by calculating the Spearman correlation coefficient and analyzing the correlation between computational scores and experimental protein folding stability, using a total of 13,442 sequences.

As shown in Fig. 3B, AF2 and ESMFold pLDDT achieve the highest correlation with experimental absolute folding stability $\Delta G_{\text{Unfold}}$, as low pLDDT values are typically observed in unfolded segments. Compared to AF2 predictions with the default setting of three recycling steps, predictions without recycling demonstrate an improved overall correlation, as shown in Supplementary Fig. 9. Additionally, fully masking the MSA input and averaging eight predictions with random seeds improves the Spearman correlation from 0.50 to 0.54 compared to providing the full sequence to the AF2 MSA module. ESMFold also achieves a higher correlation than the single-sequence approach by averaging eight pLDDT predictions with 50% random masking of the input sequence, which can enhance local stability predictions[30]. We reason that for de novo designs, language model features are less important, as they primarily capture evolutionary information. Therefore, masking language model features may improve AF2 and ESMFold predictions. While recycling enables the model to refine prediction quality, it can also result in unstable or poor-quality sequences being assigned high confidence. We also evaluated the Boltz-2 model[31], a diffusion-based structure prediction method similar to AlphaFold3[32], but found it does not show improved correlation with experimental stability compared to AF2 and ESMFold pLDDT.

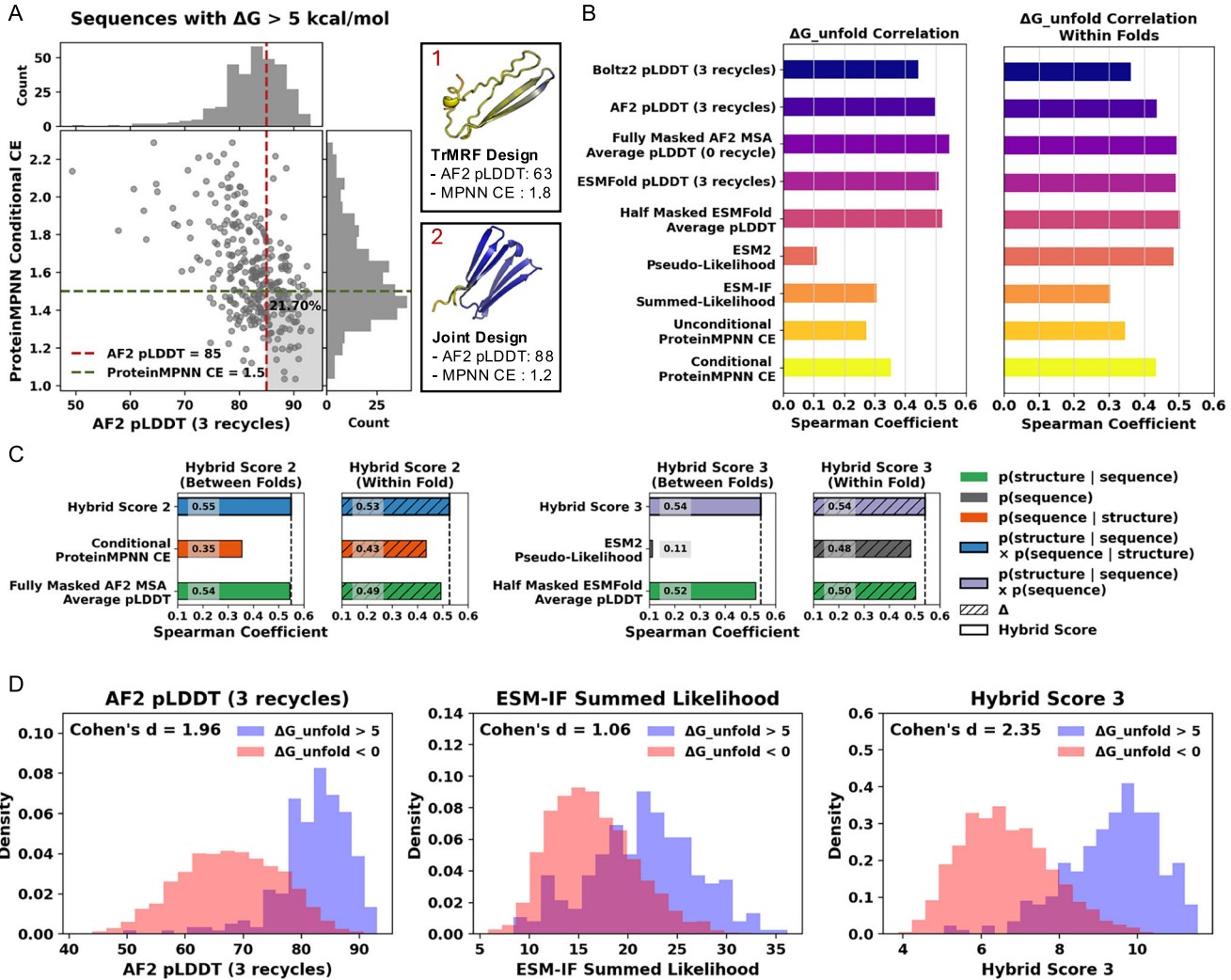

**Fig. 3 | Zero-shot evaluation of large protein models identifies that hybrid scores combining sequence and structure-Based models achieve the best correlation with actual folding stability. A** All sequences with $\Delta G_{unfold}$ > 5 kcal/mol plotted with AF2 pLDDT on the x-axis and ProteinMPNN conditional CE on the y-axis. The gray box in the bottom right highlights sequences that passed common filtering standards: AF2 pLDDT > 85 and ProteinMPNN conditional CE <1.5. (Upper) Example of a protein with low confidence but high folding stability, and (Lower) an example of a protein designed with high confidence and high stability. The structures are colored by per-residue level AF2 pLDDT value, where blue indicates high pLDDT and red indicates low pLDDT. **B** Spearman correlations between (Left) computational folding stability metrics and absolute folding stability($\Delta G_{unfold}$) across different folds, and (Right) Spearman correlations between differences in absolute folding stability within folds and differences in metrics. **C** Spearman correlations between computational folding stability metrics and $\Delta G$ for both single and hybrid scores, along with correlations: (1) between folds and (2) differences within folds sharing the same structures. **D** Distribution of stable ($\Delta G_{unfold}$ > 5 kcal/mol) and unstable designs ($\Delta G_{unfold}$ < 0 kcal/mol) based on computational scoring metrics: AF2 pLDDT (3 recycles), ESM-IF Summed Likelihood, and Hybrid score 3.

Based on Fig. 3B (left), it is clear that the ESM-2 model fails to predict the absolute folding stability compared to other models. We hypothesize that this limitation arises from the absence of a structural component in the model capable of evaluating structural stability alongside sequence stability. Instead of directly predicting folding stability, we conduct experiments to predict differences in protein folding stability across sequences derived from the same structure but generated using different models. Regarding the $\Delta G_{Unfold}$ correlation within folds (sequences generated from different models but designed from the same backbone structure), the ESM-2 pseudo-likelihood score shows a high correlation with the delta matrices (Fig. 3B, right), in contrast to the low correlation observed between different folds. By comparing the stability of sequences within the same fold, we can offset the absolute differences between the two structures and assess the stability differences arising from sequence variations.

**Combining sequence and structure models improves folding stability prediction**

We introduce three hybrid scoring methods to enhance the accuracy of folding stability predictions by combining two of the following types of scores: (1) $p$(sequence) using ESM-2 Pseudo likelihood; (2) $p$(sequence|structure) using ProteinMPNN conditional cross-entropy (CCE); and (3) $p$(structure|sequence) using either AF2 or ESMFold pLDDT.

**Hybrid Score1** $= -0.90 \cdot$ ProteinMPNN CCE
$+ 0.1 \cdot$ Half Masked ESMFold pLDDT
**Hybrid Score2** $= -0.90 \cdot$ ProteinMPNN CCE
$+ 0.10 \cdot$ Fully Masked AF2 MSA pLDDT
**Hybrid Score3** $= 0.86 \cdot$ ESM-2 Pseudo-likelihood
$+ 0.14 \cdot$ Half Masked ESMFold pLDDT

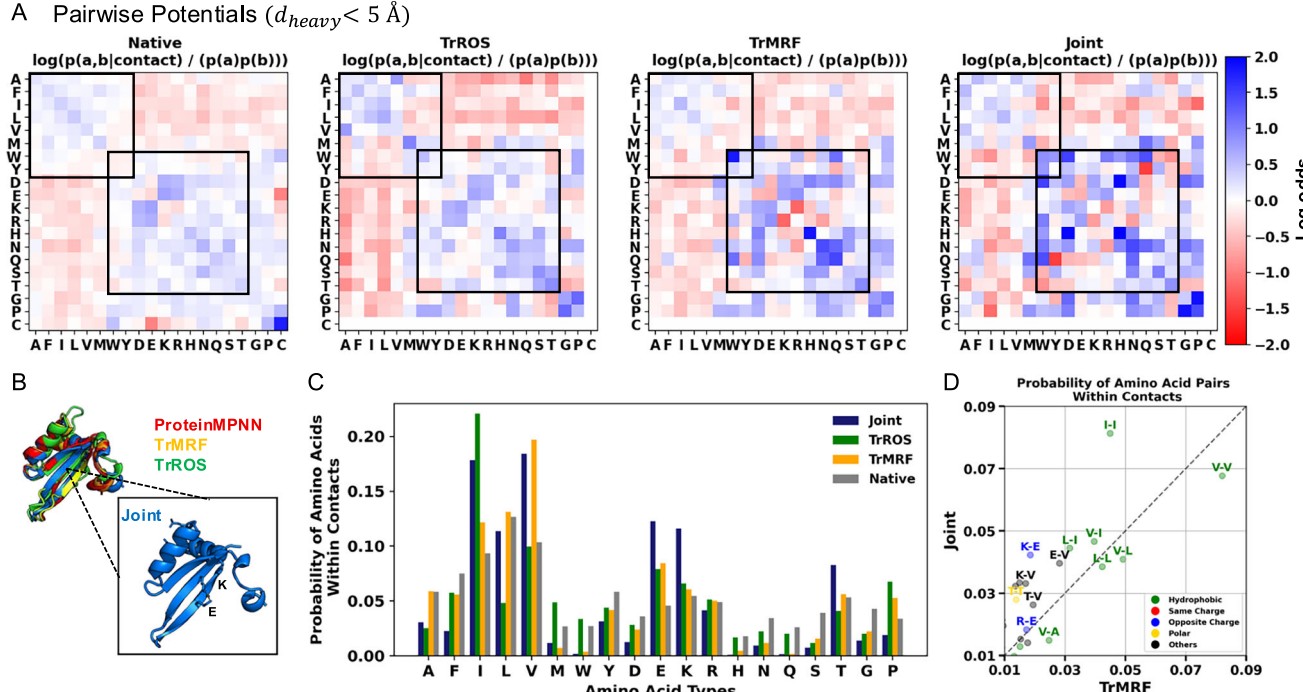

**Fig. 4 | Sitewise and pairwise analyses show that the joint model samples a higher frequency of hydrophilic interaction pairs. A** 20 × 20 amino acid pairwise potentials in contacts where distances between heavy atoms are less than 5 Å for joint model-sampled sequences. The pairwise potential is calculated as log(P(a, b | contact)) / P(a)P(b). Black boxes indicate interactions between charged and polar residues and hydrophobic residues. Blue denotes enriched interactions, while red represents depleted interactions. For the native, pairwise potentials were calculated from 200 PDB native sequences. **B** AlphaFold2-predicted structures from sequences generated by the joint, TrROS, TrMRF, and ProteinMPNN models, with a zoomed in view showing electrostatic and hydrogen bonds formed between side chains in the joint model structure. **C** Bar plots show the probability of single amino acids within contacts from the joint model, TrROS, TrMRF, and native proteins. Cysteine is excluded from the plots, as it is not used in the sequence design. **D** Scatter plots showing the probability of amino acid pairs within contacts between the Joint model and TrROS model sequences.

To determine the ratio between the two scores, we used 20% of the datasets to find the ratio that achieves the best Spearman correlation with experimental absolute stability and the hybrid score, and we obtained the final Spearman correlation tested on the remaining 80% of the dataset. Hybrid Score 2, which combines the fully masked AF2 MSA average pLDDT and ProteinMPNN CCE, and Hybrid Score 3, which combines the pseudo-likelihood of ESM-2 with a 50% masked ESMFold average pLDDT, achieve the best correlation with experimental folding stability scores (Fig. 3C). For Hybrid Score 3, optimizing both $p(structure|sequence)$ and $p(sequence)$, allows for adjusting the $p(structure|sequence)$ model, which intrinsically assumes $p(sequence) = 1$, meaning all sequences are equally likely. By incorporating $p(sequence)$, we can more accurately assess the likelihood of both the sequence and its corresponding structure. Compared to the commonly used AF2 filtering (3 recycles) and the ESM-IF summed likelihood score-presented as a zero-shot absolute protein folding stability predictor[29]-the Hybrid score more effectively distinguishes stable and unstable designs, as shown in Fig. 3D. This supports our hypothesis that combining structure- and sequence-based scores can improve the generation and prediction of stable proteins.

**Joint model designs sequences with higher frequency of hydrophilic interactions**

To understand the differences between sequences generated by the joint model and single-objective models, we focus on amino acid pairs in contact, as interactions between amino acids are crucial for maintaining both secondary and global structure. We identified these amino acid pairs by measuring the distances between heavy atoms in side chains that are less than 5 Å and separated by more than five residues in the predicted structures from AlphaFold2. As a control, we evaluated the contact pairs in 200 PDB native structures. As shown in Fig. 4A, sequences from the TrROS, TrMRF, and joint models exhibit enriched interactions of charged, polar, and hydrophobic residues, with same-charge pairs showing negative couplings[13,33]. Notably, sequences from the joint model show a higher log odds ratio for both hydrophilic residue pairs (Fig. 4A). As shown in Fig. 4B, the secondary structure of the joint model sequence is well maintained, with electrostatic and hydrogen bonds forming between side chains. Charged residues, such as lysine and glutamic acid, are sampled more frequently in joint models than in TrMRF models (Fig. 4C), with a high probability of lysine-glutamic acid pairs forming contacts (Fig. 4D), along with other hydrophilic interactions. These features may contribute to the high stability of joint model sequences, allowing them to maintain both their secondary and global structures. Additionally, analysis of residue burial patterns shows that polar interactions in joint model designs tend to be more buried than in TrROS designs, while charged pairs remain surface-exposed (Supplementary Fig. 15).

## Discussion

Our systematic analysis of (1) $P(structure|sequence)$, (2) $P(sequence|structure)$, and (3) joint optimization of both demonstrates that joint optimization implicitly models the conformational landscape. It generates the most stable proteins by simultaneously designing the sequence and adjusting the backbone to accommodate the sequence. Sequences generated by the joint model exhibit a folding stability distribution similar to those generated by $P(sequence|structure)$

models filtered with the common AF2 pLDDT standard, which also models $P$(structure|sequence). This similarity arises because filtering $P$(sequence|structure) sequences with $P$(structure|sequence) helps exclude sequences that fold into alternative states by globally exploring the conformational landscape, performing a sequential form of joint optimization. However, after applying the same AF2 pLDDT filter, joint model sequences exhibit greater stability compared to those from $P$(sequence|structure) models, suggesting that the joint model simultaneously optimizes both sequence and structure to find an optimal solution. Additionally, jointly modeling both components may help mitigate overfitting through ensemble methods, where individual models might overfit, but the joint approach reduces this issue.

We also evaluate the ability of the current models to score stability in a zero-shot setting. We compared the correlation between AF2 and ESMFold stability predictions across multiple different settings and found that, for de novo proteins, masking language model information-by masking the AF2 MSA input, masking the ESMFold input sequence, and averaging multiple outputs-helps increase the correlation with experimental stability. For AF2, decreasing recycling also prevents the model from having high confidence in adversarial sequences. Hybrid scores, derived from combining sequence and structure-based models, exhibit the highest correlation with experimental folding stability compared to any single model. Notably, the hybrid score combining (1) AF2 MSA average pLDDT and ProteinMPNN CCE, with (2) ESM2 pseudo-likelihood and half-masked ESMFold pLDDT, achieves the best results. By considering both (1) $P$(sequence|structure) and $P$(structure|sequence), and (2) $P$(sequence) and $P$(structure|sequence), we can assess the likelihood of both the sequence and its corresponding structure, thereby enhancing the hybrid score for predicting stability.

In conclusion, joint optimization enables implicit modeling of the conformational landscape and sequence design, facilitating the scoring and generation of stable proteins. Our large stability dataset offers an opportunity to evaluate different models and analyze stability across sequences sharing the same folds, allowing for the independent analysis of sequence components from the structure. Although our current work primarily evaluates mini-proteins of fewer than 80 amino acids, which may limit its applicability to larger, multi-domain proteins, we expect that this methodological analysis and dataset will lay the foundation for designing stable proteins and for optimizing and evaluating models in conformational landscape modeling.

## Methods
### TrROS, TrMRF, and joint models
TrROS integrates sequence information to estimate conservation and coevolution features calculated from the inverse covariance matrix of the input sequences. These features are then used as inputs to predict protein structures using ResNet blocks. ResNet blocks, featuring dilated convolutions and dropout layers, enable the model to capture long-range dependencies within protein sequences. Finally, the prediction head outputs 6D features, including torsion angles, distance predictions, backbone torsion angles, and omega angles. Rosetta then reconstructs the 3D structure of the protein using 6D features. For unconditional protein generation, we initialize the model with a random amino acid sequence, predict the structure, and backpropagate the gradients through the TrROS model. Based on the KL (Kullback–Leibler divergence) loss between the predicted and the background structure distance map (averaged over all protein structures), we can optimize both the amino acid sequences and the structures simultaneously. This process continues until a structure with a clear distance matrix is obtained.

TrMRF is an inverted model of TrROS more specifically, for a given structure, it predicts the conservation and coevolution features and

maximizes the pseudo-likelihood of the associated Multiple Sequence Alignment (MSA), approximating the probability of the sequence given the structure ($p$(sequence|structure)). Input features include one-hot encoded structural bins derived from the protein structure, as well as a constant sequence identity channel. For each sequence in the MSA, its average identity to the reference sequence is computed. The mean identity of the subset of sequences selected for training is encoded as a constant matrix and concatenated with the structural input channels, providing the network with a coarse measure of how similar the homologous sequences are to the reference. ResNet layers with varying dilation rates enable the model to capture features at different spatial resolutions. After the ResNet blocks, the model branches into a separate path to extract pairwise potentials and long-range dependencies. This path reshapes the output of the ResNet into a higher-dimensional tensor representing coevolution and conservation features. The model is trained by minimizing cross-entropy loss between actual and predicted sequences using 15,051 nonredundant proteins from the Protein Data Bank (≤30 % sequence identity, structures released before May 1, 2018).

For TrMRF sequence sampling, we employ an iterative optimization approach starting from random amino acid sequences. The process involves: (1) adding Gumbel noise to sequence logits ($I_{soft}$) followed by argmax operation to generate one-hot encoded sequences ($I_{hard}$), (2) computing categorical cross-entropy loss between the predicted probabilities ($I_{pred}$) and the current sequence, and (3) back-propagating gradients using a straight-through estimator ($stop\_gradient(I_{hard} - I_{soft}) + I_{soft}$) to update only the input sequence. This optimization continues until convergence, with final sequences extracted from $I_{hard}$ (Supplementary Fig. 1). During sampling, the sequence identity channel is fixed at 1.0, effectively conditioning the model to treat all sequences as fully identical to the reference, thereby biasing sequence generation toward high-identity sequences. For ProteinMPNN, sequences are sampled using the default $vanillaMPNN$ model with temperature $T = 0.1$.

The joint model combines the TrROS and TrMRF models. Without retraining the model weights, we utilize the pre-trained TrROS and TrMRF models. For sequence generation, we jointly optimize the loss from both the TrROS and TrMRF models and update the sequence accordingly. Unlike the TrMRF and ProteinMPNN models, which perform fixed backbone design by starting from the structure provided by the TrROS model, the joint model begins with the final sequence of the predicted structure. Then it uses this sequence as the initial input to predict 6D features using the TrROS model. These 6D features are given as input to the TrMRF model, which generates sequences. There are two losses in the optimization process: TrMRF loss and TrROS loss. TrMRF loss is backpropagated into the TrMRF model and the input sequence, while TrROS loss is backpropagated through the TrROS model. The optimization process continues until the loss no longer decreases.

### Data filtration
For generated sequences, we observe that some designs have hydrophobic patches that can form homo-oligomers. Therefore, even if one protein is proteolyzed, the other can bind to the fragmented protein, leading to a false signal. To computationally distinguish proteins that can potentially form homo-oligomers, we use the predicted alignment error (inter-PAE) between chains provided by AlphaFold2[34,35]. We provide two copies of proteins to AlphaFold2 and let it predict whether they form homo-oligomers using the inter-PAE value. Typically, inter-PAE > 15 indicates that the protein exists as a monomer[34].

### Sequence based zero-shot method for predicting protein stability
**ESM 2.** is a protein language model that enables the direct prediction of masked positions of various protein attributes, including structure

and function, from an individual sequence[26]. ESM-2 is trained on the UniRef database[36] by predicting 15% of the masked amino acids in input protein sequences. The pseudo-likelihood of ESM-2 approximates the probability of a sequence $P(sequence)$, and serves as a metric to assess the performance of a sequence model in representing data[26,37,38]. Since the ESM attention matrix and representations can predict contacts within the protein[39,40], we assume that pseudo-likelihood might help distinguish between stable and unstable structures. We calculate the ESM-2 pseudo-likelihood for our generated sequence $S$ of length $L$ by masking each position $s_i$ with the [MASK] token, denoted as $S_{\setminus i} := (s_1, \ldots, s_{i-1}, [MASK], s_{i+1}, \ldots, s_L)$. This calculation is performed by aggregating the conditional log probabilities $\log P_{ESM2}(s_i|S_{\setminus i})$ for each token.

$$\text{Pseudo-likelihood } (S) = \frac{1}{L}\sum_{i=1}^{L}\log P_{ESM2}(s_i|S_{\setminus i}) \qquad (1)$$

**ESMFold.** utilizes the ESM-2 language model and structure module to produce precise structure predictions for proteins[27]. ESMFold achieves accurate predictions of the structures of de novo proteins using a single sequence. It demonstrates the capability to differentiate proteins into groups that are designable and those that are undesignable[30]. ESMFold processes a protein sequence ($S$) of length ($L$), generating the predicted local distance difference test ($pLDDT$) and *distogram logits* for information about contact predictions. pLDDT is a matrix that assesses the quality of a protein structure at a local level, while distogram logits ($g(S)$) represent the structural probability distribution $p(xyz|S)$. We hypothesize that calculating the difference between predicted and actual protein structures using cross-entropy, which measures the disparity between the predicted structure distribution from the model and the true structure, could help classify protein stability by evaluating the confidence in the predicted structure. Cross-entropy, $H(F, g(S))$, is calculated by digitizing the actual structure into an $L \times L \times M$ matrix, where $M$ is the number of bins.

$$
\begin{aligned}
B_i &= [b_{i-1}, b_i), \quad i = 1, 2, \ldots, M \\
F_{ijk} &= \begin{cases} 1 & \text{if } d_{ij} \in B_k, \\ 0 & \text{otherwise}, \end{cases} \qquad (2) \\
H(F, g(S)) &= \sum_{i=1}^{L}\sum_{j=1}^{L}\sum_{k=1}^{M} -F \cdot \text{LogSoftmax}\,(g(S)_{ijk})
\end{aligned}
$$

The number of bins, denoted as $M$, is determined by dividing the total range, which is the range of pairwise contact distances, $b_M - b_1$, by the width of the bin, $h$. This can be expressed as $M = \frac{b_M - b_1}{h}$. Each bin, $B_i$, represents an interval $[b_{i-1}, b_i)$. The matrix $F$ encodes whether the value of the distance $d_{ij}$ between the $i$th and $j$th positions of $C_\beta$ belongs to a bin $B_i$. It assigns a one if it does and a 0 otherwise.

### Structure based zero-shot method for predicting protein stability

**ESM-IF.** Previous works showed that the ESM-IF model can be used to predict the absolute folding stability of proteins[28,29]. To evaluate folding stability, the authors used the output logits from the ESM-IF decoder to assess the likelihood of amino acids at each position. This likelihood is determined by applying the Softmax function to the 20 amino acid logits at each position and then summing across the length of the protein to obtain the overall folding stability.

**ProteinMPNN.** generates a sequence from protein backbone coordinates with a message-passing structure consisting of an encoder-decoder framework[15,41]. ProteinMPNN incorporates protein backbone coordinates by representing protein residues as nodes and establishing nearest neighbor edges by considering pairwise distances between backbone atoms. These input features are subsequently passed through three encoder layers and then into the decoder layers to

obtain a sequence. ProteinMPNN predicts the probability of a sequence (logits) given a structure that is in the opposite direction compared to ESMFold. We employ the ProteinMPNN cross-entropy term, $H(X, f(X))$, as a criterion for assessing the folding stability. This involves calculating the cross-entropy between the ground-truth sequence and the logits. This choice is based on the observation that ProteinMPNN can evaluate the quality of ROSETTA decoy datasets of 133 native protein structures. The cross-entropy of the protein decoy aligns well with the TM score between the decoy and the ground truth structure, as illustrated in.

$$H(X, f(X)) = \sum_{l=1}^{L}\sum_{a \in A} -X \log(f(X)_{la}) \quad \text{where } X \in \{0,1\}^{L \times A} \qquad (3)$$

$X$ is a one-hot encoded protein sequence of length $L$ and composed of $A$ different types of amino acids. Logits from the model can be generated either by conditioning on the ground truth sequence during decoding, resulting in conditional logits, or without conditioning on any sequence, termed unconditional logits. We term the cross-entropy derived from conditional logits as *conditional cross-entropy* (CCE) and that from unconditional logits as *unconditional cross-entropy* (UCE).

### Reporting summary
Further information on research design is available in the Nature Portfolio Reporting Summary linked to this article.

## Data availability
The data supporting the findings of this study, including protein sequences and experimental stability measurements, have been deposited in Zenodo under https://sandbox.zenodo.org/uploads/411658. The raw dataset containing protein model confidence scores and stability data is provided as Supplementary Data 1. Source data are provided with this paper.

## Code availability
The code used for model training, structure optimization, and stability analysis is available at https://github.com/yehlincho/Joint_Model_Stability.

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

## Acknowledgements

We thank the members of the Rocklin and Ovchinnikov labs for useful discussions. Y.C. acknowledges funding from SBS Scholarships and the Takeda Fellowship. S.O. acknowledges funding from NIH grant DP5OD026389, NSF MCB2032259, and Amgen. K.T. acknowledges funding from the Human Frontier Science Program Long-Term Fellowship and JST PRESTO Grant JPMJPR21E9. G.J.R. acknowledges funding from NIH grants DP2GM140927 and R35GM158118.

## Author contributions

Y.C., S.O., and J.D. generated the datasets and trained and validated the model. K.T. performed the protein stability assay. Y.C. and S.O. contributed to the analysis, and S.O. and G.J.R. supervised the work. Y.C. wrote the manuscript with input from all authors. All authors reviewed and approved the final version of the manuscript.

## Competing interests

The authors declare no competing interests.
