## [Transparent Peer Review file · Nature Communications]

Stable De Novo Protein Design via Joint Conformational Landscape and Sequence Optimization

Corresponding Author: Professor Sergey Ovchinnikov

Version 0:

Reviewer comments:

Reviewer #1

(Remarks to the Author)

I always appreciate seeing different approaches benchmarked and compared. Overall, this study addressed an important question, however – and the authors themselves state it similarly, it is neither a completely new idea or concept, nor a major methodological step forward. Optimizing for sequence and structure simultaneously (jointly) definitely seems better. However, I wouldn't call it definitively superior, as filtering with AlphaFold2 achieves similar outcomes – although applying the filter to the joint method's results is even better. Also, this has not systematically been shown alongside the reported data as strategy, so a comparison is not possible. This also seems to be the major limitation of the study to me – the study proves an important point, but it is more of theoretical importance, and does not provide major new innovation or advances the used methods.

The authors conducted extensive analysis and attempted to find the best correlation with experimental data. The hybrid scores are interesting, though it's not clear to me how they determined the weightings (perhaps through trial and error). It's noteworthy, however, that the correlation for different models is only moderate (around 0.5 for the best performers).

Points of Concern:

The authors state in their very first paragraph stability as the major design goal – which is true, but function is the other important aspect. And often sacrificed when designing for function.

The paper assumes TrMRF is a known sequence design method, but I couldn't find a source for it (and they don't provide one). I'm also unsure what the abbreviation stands for. While the methods section explains it as an inversion of TrROS, its origin is unclear: Did they develop it? How was it trained (they mention using a pre-trained version)?

TrROS (which seems to be TrRosetta, an abbreviation one has to deduce) generates structures starting from random sequences. It runs until a fixed distance matrix is achieved (they quote: "optimize both the amino acid sequences and the structures simultaneously"). My question is: if the output of this step is already at a loss minimum for a given sequence (as I understand it), how does applying the joint model subsequently (they quote: "the joint model begins with the final sequence of the predicted structure") generate a new sequence? How different are the final sequences from the initial TrROS step compared to those from the joint model, which starts with that final structure and uses the inverted TrMRF? I don't understand how a new sequence can be generated from an already optimized structure-sequence pair – perhaps I'm missing something.

The authors state in the very last sentence of the discussion that they only generated mini-proteins with a length (L) of less than 80 amino acids. I believe this is crucial information that should definitely be mentioned alongside their results (it's not found elsewhere). Knowing this context is important for interpreting the pLDDT scores (from AF2 or ESMFold) for the 20,000 generated/predicted/filtered sequences.

Regarding Figure 2: In my view, the density plot for the joint model in Panel B (blue) should be identical to that of the joint model without filtering in Panel C (light blue – also, Panel C is missing its label). The underlying data should be the same. However, Panel B shows small, distinct "shoulders" on both the left and right sides of the distribution that are absent in Panel C, with the left shoulder being particularly diminished. Why is this? Are these different datasets?

In figure 3b, the authors highlight half masked ESMFolds average pLDDT as the best method – overall it is slightly better – the differences seem to be not totally different from fully masked AF2 MSA average pLDDT and ESMFold pLDDT 3 recycles. I'm not very familiar with the experimental cDNA proteolysis assay, but I'm confused as to why they set the K50 value for the folded state (θ) to be the same for all randomly generated sequences (and what specific value did they use?). Does this make sense? They justify this by arguing that cleavage likely only occurs in "constant regions." Which constant regions are being referred to, given that the sequences were randomly generated? Do they mean regions that are conserved across a set of template sequences used for defining a fold (e.g., if they refer to "all 4 sequences for a Fold" in some context not fully clear

from this snippet)?

In the methods section, the experimental methods are not reported but in the Supplement – that seems to be weighting in on the computation.

The Github link leads me to an error.

Minor Points / Questions:

What is the origin of the ProteinMPNN filter threshold of 1.5 for Cross-Entropy (CE)? Notably, CE is only defined as an abbreviation in the supplementary information. How was it calculated?

Regarding sampling: They state that one sequence was generated per structure. However, for ProteinMPNN and TrMRF, there's no description of the sampling process. This is significant because parameters like sampling temperature and the specific model used (e.g., for MPNN) can drastically affect the outcome, especially when generating only a single sequence. To support the claim that the joint model doesn't alter the backbone, they use a pLDDT cutoff of >70 when calculating RMSD between pairs. However, elsewhere in the paper, they consistently use a pLDDT cutoff of >85. Why the discrepancy?

What is the success rate reported for the joint model? The first experimental section provides success percentages for TrROS and other methods, but not explicitly for the joint model.

In Supplementary Figure S9, there's a white bar on the right that obscures parts of the data.

The statement in the first results section, "The structures generated by TrROS are diverse, with an average TM score of 0.4 between runs," seems questionable. A TM score of 0.4 suggests that some structures might already share the same or a similar fold, implying less diversity than stated. The value in Supplementary Figure S4 (assuming "Supp Fig S14" refers to S4) might represent a different average.

Figure 4 Panel B seems redundant. Without the accompanying text, its added value isn't clear, especially since the textual claim (Quote: "secondary structure of the joint model sequence is better maintained compared to those from other single models,") isn't readily apparent from the graphic itself.

Is a 100% masked MSA equivalent to using a single sequence input mode

Some more stylistic points:

The affiliations are incomplete.

The abstract sounds very technical - for a more general readership omitting strictly technical terms might improve readability

(Remarks on code availability)

I ran into an 404 error and could actually not access the code.

Reviewer #2

(Remarks to the Author)

AI-based methods have been used recently, by a variety of labs, to create well folded de novo proteins, but there are no studies that have experimentally compared different AI-based design approaches on a large set of sequences. In this paper, thousands of sequences are generated with alternative design protocols and then experimentally tested. The design methods either focus on optimizing a sequence for a given structure ($P(\text{seq}|\text{struc})$), searching for sequences that are predicted to adopt a single structure with high confidence ($P(\text{struc}|\text{seq})$), or joint optimization of both the sequence and structure. With the methods employed here, the results show that joint optimization more consistently produces highly stable proteins. It is gratifying to see the approach that seems like it should be more effective, is actually more effective.

Simultaneous optimization of sequence and structure should allow for more tightly packed cores, more ideal hydrogen bonds, and structures with little backbone and side chain strain. Iterative optimization of structure and sequence has been used effectively with older physical models (such as Rosetta), and there are a few studies with AI methods that have had success by iterating between AF (or similar) and sequence optimization with ProteinMPNN (or similar). This study convincingly shows that joint optimization is a useful approach, and this paper will be of strong interest to the protein design field. The paper should be appropriate for publication in Nature Communications if most of the following points can be addressed.

- Please indicate the secondary structure content of the designs. For instance, what fraction are all helical, all beta, or mixed?

- The last sentence of the second paragraph of the results says, "confirming that continuous hallucination does not alter the structure". The point being made here is that joint optimization is not changing the fold of the proteins, but presumably small perturbations to the structure are allowing lower energy sequence/structure pairs to be identified. Consistent with this thought, the distributions for the joint optimization in figure SI 3A are shifted to the right (i.e. slighter higher RMSDs). As currently written, this section of the text doesn't acknowledge that small perturbations to the structure may be a key reason why joint optimization has the best performance.

- The authors point out that a large fraction of the designs with high measured stability (> 5 kcal/mol) have moderate AF2 confidence scores. For instance, TRMF design 1 in figure 3A. It may be important to check the stability of some of these proteins with an alternative technique besides protease resistance on the yeast cell surface. Rocklin et al have validated their method for measuring stability in previous studies, but this type of validation is not part of this study. Purifying 10-20 of these low confidence/high stability designs and measuring their stability with CD (or similar) could help establish if some designs are escaping proteolysis for reasons other than high free energy of unfolding.

- The authors point out that the joint model creates sequences with a higher frequency of hydrophilic interactions. I'm curious what structural environment these mutations are located in. Are there just more polar interactions on the protein surface, or does the joint model also create more buried polar interactions.? How does the fraction of buried polar interactions in the stable designs compare to native proteins?

- It would be nice to have a structural or physical understanding of what the joint model is doing better. If you score the different models with a physical force field, such as Rosetta (probably need to energy minimize first), is there anything that stands out. Do the joint distribution models have lower Rosetta energies (or specific energy terms that are lower?). Do the

models have similar packing metrics? It might be best to generate these models with AF3. I would also be interesting to see how well AF3 does at predicting which sequences will be most stable.

(Remarks on code availability)

Version 1:

Reviewer comments:

Reviewer #2

(Remarks to the Author)

The authors have addressed my comments and questions with this revision. This work highlights important differences between computational methods for protein design and is appropriate for publication in Nature Communications.

The authors changes to the manuscript address all the comments and questions of reviewer #1, and my opinion is that the paper is now ready for publication.

(Remarks on code availability)

REVIEWER COMMENTS

Reviewer #1 (Remarks to the Author):

I always appreciate seeing different approaches benchmarked and compared. Overall, this study addressed an important question, however – and the authors themselves state it similarly, it is neither a completely new idea or concept, nor a major methodological step forward. Optimizing for sequence and structure simultaneously (jointly) definitely seems better. However, I wouldn't call it definitively superior, as filtering with AlphaFold2 achieves similar outcomes – although applying the filter to the joint method's results is even better. Also, this has not systematically been shown alongside the reported data as strategy, so a comparison is not possible. This also seems to be the major limitation of the study to me – the study proves an important point, but it is more of theoretical importance, and does not provide major new innovation or advances the used methods.

The authors conducted extensive analysis and attempted to find the best correlation with experimental data. The hybrid scores are interesting, though it's not clear to me how they determined the weightings (perhaps through trial and error). It's noteworthy, however, that the correlation for different models is only moderate (around 0.5 for the best performers).

Points of Concern:

- 1. The authors state in their very first paragraph stability as the major design goal – which is true, but function is the other important aspect. And often sacrificed when designing for function. The paper assumes TrMRF is a known sequence design method, but I couldn't find a source for it (and they don't provide one). I'm also unsure what the abbreviation stands for. While the methods section explains it as an inversion of TrROS, its origin is unclear: Did they develop it? How was it trained (they mention using a pre-trained version)?**

- In this paper, we newly developed TrMRF (transform-restrained Markov Random Field) by inverting the TrROS (transform-restrained Rosetta) model

$$W, b = \text{TrMRF}(\text{structure})$$
$$\text{pred_sequences} = \text{softmax}(\text{sequences}@W + b)$$

- Input features include one-hot encoded structural bins derived from the protein structure, as well as a constant sequence identity channel. For each sequence in the multiple sequence alignment (MSA), its average identity to the reference sequence is computed. The mean identity of the subset of sequences selected for training is encoded as an LxL constant matrix and concatenated with the structural input channels. This provides the network with a coarse measure of how similar the homologous sequences are to the reference.
- The output of TrMRF consists of the matrices W and b, which represent the coevolution and conservation matrix from the input protein structure. The predicted sequences are obtained by applying a softmax function to the product of the input sequences and W, followed by adding b. The loss is calculated using cross-entropy between the actual sequences and the predicted sequences. The network is trained by minimizing this cross-entropy loss. The model is trained by minimizing the cross-entropy loss between the actual and predicted sequences. The training dataset comprises 15,051 proteins from the Protein Data Bank, curated to be nonredundant at a 30% sequence identity threshold and restricted to structures released before May 1, 2018.
- During sampling, the sequence identity channel can be set to 1.0, effectively telling the model to behave as if all sequences were maximally identical to the reference, biasing it toward generating high-identity sequences
- Even though TrMRF now has been replaced by many other sequence design models, we still think the dataset generated from these models is meaningful, because the we directly compared the sequences that

designed for same structure and measured absolute stability that can compare the difference between sequences.

- 2. *TrROS (which seems to be TrRosetta, an abbreviation one has to deduce) generates structures starting from random sequences. It runs until a fixed distance matrix is achieved (they quote: "optimize both the amino acid sequences and the structures simultaneously"). My question is: if the output of this step is already at a loss minimum for a given sequence (as I understand it), how does applying the joint model subsequently (they quote: "the joint model begins with the final sequence of the predicted structure") generate a new sequence?***

- From the perspective of a structure prediction model, it aims to maximize the energy difference between the folded state and the lowest-energy alternative states, rather than directly lowering the absolute energy of the folded state. Hallucinating structure models, once they identify a confident structure–sequence pair, focus on maximizing the probability of that pair compared to other possible alternatives. In contrast, the TrMRF model (structure → sequence), which is trained by minimizing the cross-entropy (CCE) loss of sequences, learns the absolute differences and attempts to minimize the energy for a given structure. In addition, TrRosetta, an example of a P(structure|sequence) model, is effective at disfavoring alternative conformational states but is relatively limited in its ability to reduce the absolute energy of a design compared to energy-based models such as Rosetta (norn2021protein). In response to this comment, we have incorporated additional explanation in the first paragraph of Page 2.

- 3. *How different are the final sequences from the initial TrROS step compared to those from the joint model, which starts with that final structure and uses the inverted TrMRF? I don't understand how a new sequence can be generated from an already optimized structure-sequence pair – perhaps I'm missing something.***

- In the Figure SI 3(B). The average sequence similarity between TrROS and the joint model is 0.402, indicating that further optimization, especially when combined with the TrMRF model, can generate novel sequences that differ significantly from those produced by TrROS alone.
- Inverting a structure prediction model to design a protein sequence does not necessarily optimize the sequence once it achieves a confident fold among other possible conformations, because there is no driving force (gradient) to further minimize the conformational energy.
- This also supports the observation that sequence design models can often generate multiple diverse sequences that fold into the same target structure, as demonstrated in previous studies (dauparas2022robust, sumida2024improving)
- Additionally, there is a possibility that the model may produce a blurry distogram when multiple possible states are mixed, whereas a clear distogram is obtained when one state is significantly more probable. However, a clear distogram does not necessarily indicate that the corresponding state has the lowest energy.
- In response to this comment, we have incorporated additional explanation in the fourth paragraph of Page 2.

- 4. *The authors state in the very last sentence of the discussion that they only generated mini-proteins with a length (L) of less than 80 amino acids. I believe this is crucial information that should definitely be mentioned alongside their results (it's not found elsewhere). Knowing this context is important for interpreting the pLDDT scores (from AF2 or ESMFold) for the 20,000 generated/predicted/filtered sequences.***

- I completely agree with your point. I will add this information to the main result part, so make it cautious that it's not generalizable across all the different protein types
- In response to this comment, we added "To ensure compatibility with downstream experimental validation, we specifically designed mini-proteins containing fewer than 80 residues, allowing stability measurements via the cDNA display method within a high-confidence range" in the third paragraph of Page 2.

5. **Regarding Figure 2: In my view, the density plot for the joint model in Panel B (blue) should be identical to that of the joint model without filtering in Panel C (light blue – also, Panel C is missing its label). The underlying data should be the same. However, Panel B shows small, distinct "shoulders" on both the left and right sides of the distribution that are absent in Panel C, with the left shoulder being particularly diminished. Why is this? Are these different datasets?**

- Thank you for pointing out my mistake. It's the same dataset, but I had used a slightly different bandwidth when drawing the Seaborn KDE plot. I've now adjusted the figure accordingly, as shown below.

6. **In figure 3b, the authors highlight half masked ESMFolds average pLDDT as the best method – overall it is slightly better – the differences seem to be not totally different from fully masked AF2 MSA average pLDDT and ESMFold pLDDT 3 recycles.**

- I got your point and agree that these different conditions do not lead to major changes. However, as shown in Supplementary Figure SI 9, even for these simple de novo proteins, confidence scores can still vary depending on the settings. For example, changing the number of recycles from 6 to 0 shifts the confidence score from 0.48 to 0.52. Similarly, allowing the full sequence to appear as the first line of the MSA versus completely masking it changes the confidence from 0.52 to 0.54.
- These results provide insight that, for de novo proteins, the confidence module could potentially be adjusted differently to improve interpretability. Simply forcing the model to output high confidence scores does not necessarily correlate with the actual fitness of the protein in real-world settings.
- While we can tweak the structure prediction model, in general, using (1) a masked MSA and (2) disabling recycling tends to produce the most reliable results particularly in the case of de novo proteins.

7. **I'm not very familiar with the experimental cDNA proteolysis assay, but I'm confused as to why they set the K50 value for the folded state (ΔG_{fold}) to be the same for all randomly generated sequences (and what specific value did they use?). Does this make sense? They justify this by arguing that cleavage likely only occurs in "constant regions." Which constant regions are being referred to, given that the sequences were randomly generated? Do they mean regions that are conserved across a set of template sequences used for defining a fold (e.g., if they refer to "all 4 sequences for a Fold" in some context not fully clear from this snippet)?**

- Yes, we assume that if a protein is in the folded state, it won't be cleaved by a protease. However, we cannot prove this, since there are no proteins that exist solely in the folded state. Nevertheless, based on this assumption, we demonstrated that we can back-calculate reasonable ΔG values, which are consistent with those quantified by other methods in the previous Mega-scale study (*tsuboyama2023mega*)
- To clarify, since we assume that folded-state cleavage occurs only in the constant regions of the construct such as the N-terminal PA tag, which is used to pull down the proteins we apply the same K50, folded value across all sequences. Proteins are normally cleaved in the unfolded (U) state but can also be cleaved in the folded (F) state via cleavage of the PA tag.
- In response to this comment, we added explanation to the third paragraph on supplementary page 1.

8. In the methods section, the experimental methods are not reported but in the Supplement – that seems to be weighting in on the computation.

- Thank you for pointing that out. You can find the relevant information on Page 7, in the paragraph titled “Combining Sequence and Structure Models Improves Folding Stability Prediction.”
- To clarify further in response to your question: to determine the optimal weighting between the two scores, we used 20% of the dataset as a training set to identify the ratio that maximizes the Spearman correlation between the hybrid score and the experimental absolute stability (ΔG).
- Specifically, we scanned coefficient ratios from 0 to 1 in increments of 0.00625, applying a weight of x to one score and $(1-x)$ to the other. For each ratio, we computed the hybrid score and determined the value of x that produced the highest Spearman correlation with ΔG on the training set. This optimal ratio was then applied to the remaining 80% of the dataset (test set), where we reported the Spearman correlations between the experimental ΔG values and their corresponding optimized hybrid scores.

The Github link leads me to an error.

- We have added the GitHub link to the manuscript. However, the repository was not initially public during the revision process. To ensure accessibility, we have downloaded and attached the full GitHub codebase with the submission, and have also made it public so that reviewers can access and review the code in its current state

Minor Points / Questions:

1. What is the origin of the ProteinMPNN filter threshold of 1.5 for Cross-Entropy (CE)? Notably, CE is only defined as an abbreviation in the supplementary information. How was it calculated?

- Cross-entropy is computed as follows: for each residue position, the model predicts a probability distribution over the 20 standard amino acids. These raw predictions are passed through a softmax function to obtain normalized probabilities. The probability assigned to the ground-truth amino acid at each position is extracted, and the natural logarithm of each is taken. These log-probabilities are then summed and averaged across the entire sequence. The final cross-entropy value is the negative of this average, representing how well the model's predicted distribution matches the true sequence.
- In the ProteinMPNN test dataset, perplexity defined as the exponentiated categorical cross-entropy loss per residue is measured. The average perplexity is around 4.5 for the vanillaMPNN model with noise set to 0.2, meaning the average cross-entropy loss is approximately 1.5 (since $e^{1.5} \approx 4.5$). We use this average value as a threshold.
- We have added this explanation to Figure 5, paragraph 4.

2. Regarding sampling: They state that one sequence was generated per structure. However, for ProteinMPNN and TrMRF, there's no description of the sampling process. This is significant because parameters like sampling temperature and the specific model used (e.g., for MPNN) can drastically affect the outcome, especially when generating only a single sequence.

- For TrMRF sequence sampling, the optimization process starts with a random amino acid sequence. A categorical cross-entropy loss is calculated between this sequence and the model's predicted probabilities, with updates applied only to the input sequence. As illustrated in Supplementary Figure SI-XX, Gumbel

noise is added to the sequence logits (referred to as I_{soft}), and an argmax operation is used to generate a one-hot encoded sequence (I_{hard}). The model's conservation and coevolution features are then used to produce the predicted probabilities (I_{pred}). The loss between I_{pred} and I_{hard} is computed and backpropagated to I_{hard} using a straight-through estimator, implemented as: $stop_gradient(I_{hard} - I_{soft}) + I_{soft}$. This optimization process repeats until convergence, and the final output sequence is taken from I_{hard} .

- For ProteinMPNN, sequences are sampled using the default *vanillaMPNN* model with a temperature setting of 0.1.
- We added this detailed explanation to the methods section on page 9.

3. To support the claim that the joint model doesn't alter the backbone, they use a pLDDT cutoff of >70 when calculating RMSD between pairs. However, elsewhere in the paper, they consistently use a pLDDT cutoff of >85. Why the discrepancy?

- Thank you for pointing out the differences. This apparent discrepancy is intentional and reflects the different objectives of the analyses. A pLDDT threshold of 85 is widely used in protein design to rigorously exclude low-confidence or poorly modeled regions, ensuring that only high-quality predictions are considered (pillai2024novo, rettie2025cyclic). Therefore, we use the >85 pLDDT threshold in our main results to demonstrate that the joint model performs well even under strict design-quality constraints.
- In contrast, the appendix employs a more lenient cutoff of >70 when calculating RMSD. This difference reflects the distinct goals of each analysis: the appendix aims to capture overall structural differences between designs rather than focusing exclusively on high-confidence regions. According to the AlphaFold paper and official FAQ, regions with pLDDT scores between 50 and 70 are considered low confidence, while a cutoff of >70 generally corresponds to a correctly predicted backbone (tunyasuvunakool2021highly).
- We added the following sentences to the paper:
- Among all sequences with high folding stability ($\Delta G > 5$), only 21.7% passed both the AF2 confidence filter (pLDDT > 85) and the ProteinMPNN filter (categorical cross-entropy < 1.5). These thresholds reflect commonly used design criteria, where a cross-entropy value below 1.5 corresponds to average model perplexity, and a pLDDT score above 85 is typically used to exclude low-confidence regions (pillai2024novo, rettie2025cyclic). We have added this explanation to Figure 5, paragraph 4.

4. What is the success rate reported for the joint model? The first experimental section provides success percentages for TrROS and other methods, but not explicitly for the joint model.

- Thank you for your question. The success rate for the joint model is indeed reported in the manuscript. Specifically, Figure 2A presents a direct comparison of the joint model's success rate alongside those of TrROS, TrMRF, and ProteinMPNN. Additionally, detailed individual results can be found in Supplementary Figure 7. We aimed to highlight the joint model's performance relative to these other methods in the main figure. Please let us know if you would like us to clarify or emphasize this further in the text.

5. In Supplementary Figure S9, there's a white bar on the right that obscures parts of the data.

- Thank you for pointing that out. We have removed the white bar from Supplementary Figure S9 to ensure the data is fully visible.

6. The statement in the first results section, "The structures generated by TrROS are diverse, with an average TM score of 0.4 between runs," seems questionable. A TM score of 0.4 suggests that some structures might already share the same or a similar fold, implying less diversity than stated. The value in Supplementary Figure S4 (assuming "Supp Fig S14" refers to S4) might represent a different average.

- Thank you for pointing that out. We apologize for the mistake. The average TMscore is 0.19, and only 0.37% of sequences have similar folds with a TMscore greater than 0.4.

7. Figure 4 Panel B seems redundant. Without the accompanying text, its added value isn't clear, especially since the textual claim (Quote: "secondary structure of the joint model sequence is better

maintained compared to those from other single models,") isn't readily apparent from the graphic itself.

- We agree that this point is difficult to convey from the graphic alone. Therefore, we have revised the sentence to: "As shown in Figure 4B, the secondary structure of the joint model sequence is well maintained, with electrostatic and hydrogen bonds forming between side chains."

8. Is a 100% masked MSA equivalent to using a single sequence input mode

- No, a 100% masked MSA is not equivalent to using the single sequence input mode. In single sequence mode, both the residue information and the MSA consist of just one sequence. In contrast, when we use a 100% masked MSA, we effectively remove any dependency on the MSA module by masking all positions, but the input format still differs from true single sequence mode.

Some more stylistic points:

1. The affiliations are incomplete.

- *Thank you for the suggestion. We have now completed the affiliations accordingly.*
- Massachusetts Institute of Technology, Cambridge, MA, USA
- Department of Pharmacology & Center for Synthetic Biology, Northwestern University Feinberg School of Medicine, Chicago, IL, USA
- Institute of Industrial Science, The University of Tokyo, Tokyo, Japan
Department of Biochemistry, University of Washington, Seattle, WA, USA

2. The abstract sounds very technical - for a more general readership omitting strictly technical terms might improve readability

- Instead of using probability terms like $P(\text{sequence}|\text{structure})$ or $P(\text{structure}|\text{sequence})$, we agree that it is better to generalize the terminology to improve readability.
- In this study, we systematically analyze a joint modeling approach that simultaneously considers how likely a sequence is to fold into a target structure and how likely a structure is to give rise to that sequence.
- We further investigate features of sequences generated from the joint model and find that they exhibit higher frequencies of hydrophilic interactions, which may help maintain both secondary structure registry and pairing, that structure-to-sequence models alone often fail to capture.

Reviewer #2 (Remarks to the Author):

AI-based methods have been used recently, by a variety of labs, to create well folded de novo proteins, but there are no studies that have experimentally compared different AI-based design approaches on a large set of sequences. In this paper, thousands of sequences are generated with alternative design protocols and then experimentally tested. The design methods either focus on optimizing a sequence for a given structure ($P(\text{seq}|\text{struc})$), searching for sequences that are predicted to adopt a single structure with high confidence ($P(\text{struc}|\text{seq})$), or joint optimization of both the sequence and structure. With the methods employed here, the results show that joint optimization more consistently produces highly stable proteins. It is gratifying to see the approach that seems like it should be more effective, is actually more effective. Simultaneous optimization of sequence and structure should allow for more tightly packed cores, more ideal hydrogen bonds, and structures with little backbone and side chain strain. Iterative optimization of structure and sequence has been used effectively with older physical models (such as Rosetta), and there are a few studies with AI methods that have had success by iterating between AF (or similar) and sequence optimization with ProteinMPNN (or similar). This study convincingly shows that joint optimization is a useful approach, and this paper will be of strong interest to the protein design field. The paper should be appropriate for publication in Nature Communications if most of the following points can be addressed.

1. Please indicate the secondary structure content of the designs. For instance, what fraction are all helical, all beta, or mixed?

- We agree that more detailed information is needed. We define the classification as follows: If helix content is over 40% and sheet content is under 20%, it is classified as 'alpha.' If sheet content is over 40% and helix

content is under 20%, it is classified as 'beta.' If both helix and sheet contents exceed 20%, it is classified as 'alpha_beta.' Otherwise, it is classified as 'other.'

- As shown in Supplementary Figure 5, 54% are classified as beta, 32% as alpha + beta and 8% as alpha, indicating that many structures contain beta secondary structures.

2. The last sentence of the second paragraph of the results says, “confirming that continuous hallucination does not alter the structure”. The point being made here is that *joint optimization is not changing the fold of the proteins, but presumably small perturbations to the structure are allowing lower energy sequence/structure pairs to be identified. Consistent with this thought, the distributions for the joint optimization in figure SI 3A are shifted to the right (i.e. slighter higher RMSDs). As currently written, this section of the text doesn’t acknowledge that small perturbations to the structure may be a key reason why joint optimization has the best performance.*

- We fully agree with your point and have added additional explanation in the paper.
- We have added the following clarification: On page 2, in the final paragraph under the subtitle 'Generating diverse protein sets by jointly optimizing sequence and structure,' we clarify that while structural differences between the joint model and others are comparable to inter-model variability (Supplementary Figure 3), the joint model shows slightly right-shifted RMSD distributions (Supplementary Figure 3A), indicating modest structural perturbations

3. *The authors point out that a large fraction of the designs with high measured stability (> 5 kcal/mol) have moderate AF2 confidence scores. For instance, TRMF design 1 in figure 3A. It may be important to check the stability of some of these proteins with an alternative technique besides protease resistance on the yeast cell surface. Rocklin et al have validated their method for measuring stability in previous studies, but this type of validation is not part of this study. Purifying 10-20 of these low confidence/high stability designs and measuring their stability with CD (or similar) could help establish if some designs are escaping proteolysis for reasons other than high free energy of unfolding*

- We thank the reviewer for this important observation. We would like to clarify that, in our study, protease stability was measured using a protease digestion assay within the cDNA display system, rather than yeast surface display. This high-throughput method has been previously characterized and shown to correlate well with thermodynamic stability determined by lower-throughput techniques such as circular dichroism (CD) spectroscopy and chemical denaturation in detergent-based assays. While we acknowledge that any high-throughput assay contains some degree of noise, prior benchmarking indicates that this method provides a reliable proxy for folding free energy (ΔG), particularly when comparing relative stabilities across large numbers of designs.

The authors point out that the joint model creates sequences with a higher frequency of hydrophilic interactions. I'm curious what structural environment these mutations are located in. Are there just more polar interactions on the protein surface, or does the joint model also create more buried polar interactions.? How does the fraction of buried polar interactions in the stable designs compare to native proteins?

- Additionally, we evaluated whether these favorable interactions are more exposed or more buried in joint designs compared to those from other models. As shown in Supplementary Figure S, polar interaction pairs in the joint model tend to be relatively more buried compared to those in TrROS (i.e., they have lower normalized SASA), while charged residue pairs are more exposed on the surface. For certain amino acid pairs, such as asparagine pairing with serine, glutamine, and threonine, the joint model shows a higher proportion of buried pairs (normalized SASA < 0.2). ProteinMPNN exhibits higher normalized SASA across most residue pairs, indicating a greater proportion of surface-exposed pairs compared to the other models.
- We added this information to Supplementary Figure 15.

4. It would be nice to have a structural or physical understanding of what the joint model is doing better. If you score the different models with a physical force field, such as Rosetta (probably need to energy minimize first), is there anything that stands out. Do the joint distribution models have lower Rosetta energies (or specific energy terms that are lower?). Do the models have similar packing metrics? It might be best to generate these models with AF3. I would also be interesting to see how well AF3 does at predicting which sequences will be most stable.

- We additionally evaluated the AF3-like model Boltz2 using its default settings. We computed the correlation between stability and Boltz2 pLDDT score. As shown in the figure with Boltz2 scores, it achieves lower correlation than AF2 and ESMFold for many designs. When we compare AF2 and Boltz2 using the same recycle setting (3), on average, AF2 achieves higher pLDDT scores for the same designs. We have revised Figure 3 and added this information to the end of page 5.